# Effects of Active Spinal Orthosis on Fatty Infiltration in Paraspinal Muscles in Kyphotic Women with Osteoporotic Vertebral Fracture—Sub-Analysis of a Randomized Controlled Trial

**DOI:** 10.3390/healthcare13111262

**Published:** 2025-05-27

**Authors:** Marco Hiller, Matthias Kohl, Oliver Chaudry, Klaus Engelke, Simon von Stengel, Wolfgang Kemmler

**Affiliations:** 1Institute of Radiology, University Hospital Erlangen, 91054 Erlangen, Germany; marco.hiller@fau.de (M.H.); oliver.chaudry@fau.de (O.C.); simon.von.stengel@fau.de (S.v.S.); 2Department of Health, Medical and Life Sciences, University of Furtwangen, 78056 Schwenningen, Germany; matthias.kohl@hs-furtwangen.de; 3Department of Medicine III, University Hospital Erlangen, 91054 Erlangen, Germany; klaus.engelke@fau.de

**Keywords:** active orthosis, fat fraction, intermuscular adipose tissue, magnetic resonance imaging, muscle volume, postmenopausal women

## Abstract

**Background/Objectives**: Fatty infiltration of muscle is a predictor of degeneration. The present study determined the effect of an active spinal orthosis on muscle quality as determined by fatty infiltration in paraspinal muscles in older women with vertebral fractures and kyphosis. **Methods**: Twenty-one community-dwelling women ≥65 years with chronic back pain and vertebral fractures ≥3 months were randomly allocated to a group which wore the Spinomed active orthoses 2 × 2–3 h/d for 16 weeks (SOG: *n* = 11) or an untreated control group (CG: *n* = 10). Outcomes of the present study were parameters related to fatty infiltration of the musculi erector spinae and psoas major as determined by Magnetic Resonance Imaging (MRI). We applied a per protocol analysis; data were consistently adjusted for baseline values applying an ANCOVA. **Results**: Despite positive trends for all MRI parameters, no significant effects of the active spinal orthosis on fat infiltration of the musculus erector spinae were observed. Significant positive effects were, however, determined for musculus psoas major intra-fascial volume (*p* = 0.021; d’: 1.18) and muscle tissue volume (*p* = 0.001; d’: 1.80). No further significant effects on m. psoas major intra-fascial or muscle tissue average fat fraction or m. psoas major intramuscular adipose tissue volume were assessed. Of importance, no changes in variables that might have confounded the present result were reported. **Conclusions**: In line with recent exercise studies, the present high-volume, low-intensity back-strengthening intervention, induced by an active spinal orthosis, failed to generate significant effects on MRI measures of the m. erector spinae. On the other hand, significant effects on m. psoas major hypertrophy, albeit not fatty muscle infiltration, were determined. This new and unexpected finding should be confirmed by future studies.

## 1. Introduction

Active spinal orthoses are validated components of conservative therapy for vertebral body fractures [1,2,3]. There is considerable evidence that the positive impact on back pain intensity, physical function and kyphosis angle is induced by the uprighting and strengthening effect of the active spinal orthosis [3,4,5], provided by its biofeedback system [2,4]. While on the whole large effects of active orthosis on trunk extensor strength were reported (summary in [6]), the effect on muscle hypertrophy has yet to be evaluated. Reviewing the literature, few studies focus on parameters related to muscle tissue changes, i.e., changes of muscle volume or fatty muscle infiltration after physical interventions [7,8]. In summary, the studies reported a lack of significant effects of (resistance-type) exercise on fatty infiltration in paraspinal muscles, be it in people with low back pain [8] or other cohorts [7]. Nevertheless, there is some evidence for diverging effects when comparing active spinal orthosis with their high volume (4 h/d) low intensity approach [3] with traditional resistance-type exercise. In order to address muscular effects of active spinal orthosis, we monitored fatty muscle infiltration in paraspinal muscles by magnetic resonance imaging (MRI) in a group of older women with osteopenic vertebral fractures, kyphosis and back pain.

## 2. Materials and Methods

The present study used MRT data from a clinical trial [3] that focused on the effects of the Spinomed active spinal orthosis on chronic back pain in older kyphotic women with osteoporotic vertebral fracture. The study was conducted between January 2021 and January 2022 at the Institute of Medical Physics, Friedrich-Alexander-University of Erlangen-Nürnberg (FAU), Germany, as a randomized controlled, semi-blinded (i.e., outcome assessors, see below) trial (RCT) with a parallel group design. The trial was approved by the Ethics Committee (number 311-19b) and fully complied with the Helsinki Declaration [9]. After detailed information was provided, all the study participants gave their written informed consent. ClinicalTrials.gov: NCT04854629.

### 2.1. Participants

Details of the recruitment process were reported previously [3]. Briefly, 80 women were included after applying the following main eligibility criteria: (a) community-dwelling women, ≥65 years with (b) ≥1 osteoporotic vertebral fracture ≥3 months, (c) chronic back pain, (d) hyperkyphosis (≥50°), (e) no relevant medication and diseases with impact on study outcomes, (f) structurally fixed kyphosis with lack of extension ability of the thoracic spine, (g) kyphoplasty or vertebroplasty, and (h) use of spinal orthoses during the last 6 months. Stratified for back pain intensity, participants were randomly assigned to the spine orthosis group (SOG: *n* = 40) or the non-treated control group (CG: *n* = 40) [3] applying allocation concealment.

In the present sub-analysis, only participants for whom additional MRI imaging was performed were included (*n* = 21). Of importance, participants of the groups were not randomly assigned but requested to participate in the follow-up MRI assessment after the intervention period.

### 2.2. Intervention

Briefly, participants of the treatment group (SOG) were provided with the Spinomed active orthosis (Medi GmbH & Co. KG, Bayreuth, Germany). In brief, the Spinomed active orthosis is a tightly fitting bodysuit with textile traction and pressure elements, as well as a pocket at the back for inserting an aluminum back splint. The splint is customized to fit the patient’s back. This biofeedback system aims to remind patients to maintain an upright position, stimulating active straightening of the trunk muscles.

Most participants used the standard product; in a few cases. a custom-made orthosis was provided. In each case, the orthosis was individually fitted by an orthopedic technician who carefully instructed participants on handling, correct fit, and care of the orthoses. The same orthopedic technician checked proper orthosis fit after two and eight weeks. During the 16-week intervention, the SOG wore the orthosis daily. After two weeks of familiarization (2 h/day), wearing time increased to a 2–3 h/session twice a day during everyday physical activities. Between the two daily applications, the orthosis could be taken off or the splint could be removed from its back pocket. Participants recorded wearing time of the orthosis daily in specified logs, immediately after wearing the orthosis. Biweekly standardized telephone interviews monitored compliance with the wearing and study protocol, handling of the orthosis, complaints from wearing the orthosis and adverse effects. Further, both SOG and control groups were called every two weeks to determine changes in confounders (i.e., physical activity and exercise, physiotherapy, medication, nutritional supplements, diseases, other pain conditions, events with an impact on well-being).

Apart from the intended intervention, participants of both groups were asked to maintain their usual lifestyle, physical activity, physical therapy, medication, and other aspects with impact on our outcomes.

### 2.3. Study Outcome

The primary study outcome of the project was “Changes in average back pain intensity” [3]. However, in the present study, we addressed secondary study outcomes, i.e., changes in parameters related to fatty infiltration in paraspinal muscles (see below) as determined by MRI from baseline to 4-month follow-up.

### 2.4. Assessments

In order to avoid muscular fatigue prior to the isometric strength tests and to standardize body composition assessments, participants were asked to refrain from intense physical activity and exercise 48 h pre-assessment and to fast 2 h prior to the assessments. Baseline and 16-week control tests were conducted with the identical calibrated devices at the same time of day (maximum time deviation ± 90 min) by the same researchers/test assistants/radiologists (who conducted, segmented, or analyzed the (MRI) assessments) blinded for the participants’ group status (SOG or CG).

#### 2.4.1. Magnetic Resonance Imaging (MRI)

MR images of the lumbar spine were acquired using a 3T scanner (MAGNETOM Prisma-fit, Siemens Healthineers AG, Erlangen, Germany) with a body surface coil. A three-plane localizer scan was used to define the scan area, which completely covered the L2 to L4 vertebrae. An axial T1-weighted (T1w) turbo spin echo sequence was acquired: voxel size 0.4 × 0.4 × 3 mm^3^, slice gap 0 mm, matrix size 576 × 396, 36 slices per scan. Additionally, a volumetric interpolated breath-hold 6-point Dixon sequence was acquired to generate fat fraction (FF) maps [10]. In these maps, the grey-values (0–1000) of a given voxel encode the FF as 0.0% to 100.0% fat content. Dixon images were obtained with a voxel size of 0.9 × 0.9 × 3 mm^3^, a slice gap of 0 mm, and a matrix of 288 × 192 in 36 slices per scan.

The T1w images were used to segment the paraspinal muscles in six axial slices starting with the most cranial slice, covering L1, to the most caudal slice, covering L4, using a slice distance of 1.5 cm. In these slices, the left and right musculus (m.) psoas major and left and right m. erector spinae were manually contoured using ImageJ (version 1.54, the Laboratory for Optical and Computational Instrumentation, Washington, DC, USA). In a second step, the resulting segmentation masks were registered to the Dixon FF maps. Both steps were monitored and checked by an experienced reader. Apart from visual inspection, no further reliability assessments were conducted. Each of the four muscles was then divided into muscle tissue (MT) and intermuscular adipose tissue (IMAT) (muscle volume = MT volume + IMAT volume) using a subject-specific threshold. This threshold was determined as the minimum from the logarithmically scaled histogram of the FF values of the VOI, as described previously [11]. By using a subject-specific threshold over a fixed threshold, individual differences and partial volume artefacts can be better compensated for. (In our experience, this is quite important when comparing groups with larger differences in fat infiltration.) Finally, the volume and FF of each outlined muscle, volume of IMAT, and volume and FF within MT only were determined (Figure 1). Analysis was performed using MIAF muscle (Medical Image Analysis Framework, version 1.0.1, University of Erlangen).

#### 2.4.2. Maximum Isometric Trunk Strength Assessment

Maximum isometric trunk extension and flexion strength were measured in an upright standing position (0°) with flexed knees (20°) using an isometric strength testing machine (Back-Check^®^ 607, Dr. Wolff, Arnsberg, Germany). Two measurements were performed, and the better result was included in the analysis.

#### 2.4.3. Participant Characteristics and Confounding Factors

All participants [12] completed a standardized questionnaire that asked for (a) demographic parameters, (b) diseases and physical limitations, (c) pharmaceutic therapy/medication with particular regard to bone-specific drugs, analgesics, and corticosteroids, (d) dietary supplements (e.g., Vit-D, Calcium), (e) lifestyle, including physical activity, exercise, and nutrition. Likewise, a follow-up (FU) questionnaire asked for changes in pharmacologic therapy, diseases, surgery, lifestyle, and physical activity/exercise with potential effects on the study outcomes.

### 2.5. Statistical Analysis

The sample size calculation of the present project was based on a parameter (“changes in averaged back pain”) not addressed in the present contribution.

We conducted a per-protocol analysis that only included participants with full data sets. After checking normal distributions of the data, the study outcomes addressed here were analyzed by pre-post *t*-test comparisons with pooled SD. Differences in intra-group changes were analyzed by ANCOVA that adjusted for baseline differences. Categorical variables (Table 1) were addressed using the chi-square test. All tests were two-tailed, significance was accepted at *p* < 0.05. Standardized Mean Difference (SMD) was calculated according to Cohen (Cohens d’ [13]). SMDs (d’ values) ≥ 0.2, 0.5, and 0.8 represent small, medium, and large effects. The statistical analysis was performed using the R software package (version 4.4.1).

## 3. Results

Eleven participants of the SOG and ten participants of the CG with full data sets were included in the present analysis on the effect of active spinal orthosis on fatty infiltration in paraspinal muscles. Average daily wearing frequency was 1.71 ± 0.37 sessions/day. All the women used the orthosis at least daily, whilst only one woman was fully compliant with the protocol (2 × 2–3 h/d). Wearing duration per session averaged 126 ± 23 min (range 102–158 min) during the last 14 weeks of the intervention. Muscle soreness during the first weeks was reported by two participants of the SOG group. One participant with existing shoulder problems reported difficulties in inserting and removing the splint while wearing the body.

### 3.1. Baseline Characteristics

We observed no significant differences in baseline characteristics between the groups. In particular, variables that might affect intervention effects on the present outcome (e.g., back pain intensity, kyphosis angle, back pain intensity) did not vary between the groups.

### 3.2. Study Outcomes

After 4 months of intervention, we observed no significant effects (i.e., group differences for intra-group changes) or significant intra-group changes for any of the measures of fat infiltration of the m. erector spinae (Table 1), which should be considered as the region predominately addressed by the orthosis. Nevertheless, the SOG revealed consistently more favorable results (effect size: d’: 0.10 to 0.75; Table 2) compared with the control group.

On the other hand, significant effects for psoas major muscle volume (*p* = 0.021, d’: 1.18) and particularly psoas major muscle tissue volume (*p* = 0.001, d’: 1.80) were observed. Psoas major muscle fat fraction (FF) and muscle tissue FF decreased non-significantly in the SOG and increased non-significantly in the CG; however, the between-group differences were not significant (Table 2; d’: 0.32 and 0.38). No differences were observed for m. psoas major IMAT volume (Table 2, d’: 0.11).

#### Exploratory Outcomes

Maximum isometric trunk extension strength increased in the SOG (2.0 ± 1.9 N, *p* = 0.009) and the CG (0.5 ± 1.8 N, *p* = 379.), while significant group differences were not observed (*p* = 0.087). In contrast, significant effects (*p* < 0.001) were determined for maximum isometric trunk flexion (SOG: 3.4 ± 1.6 N, *p* = 0.001 vs. CG: 0.5 ± 1.6 N, *p* = 0.272).

### 3.3. Confounding Parameters

In summary, no significant between-group differences for pharmacologic therapy, diseases, injuries, physical activity/exercise, diet, or lifestyle factors that might have affected the outcomes addressed here were reported. In parallel, developing conditions or diseases with impact on fatty muscle infiltration were not observed. Further, no relevant changes of medication with potential impact on the present result were reported.

## 4. Discussion

Using data of a recent study on active spinal orthosis with kyphotic women, vertebral fractures, and back pain [3] as a vehicle, the present study retrospectively analyzed fatty infiltration of the paraspinal muscles in a subgroup with MRI data. In summary, we observed an overall trend to more favorable results in the SOG group (Table 2). Nevertheless, contrary to our expectations, we determined only a non-significant (albeit consistent for all measures) trend for positive effects of the active orthosis on measures of fatty infiltration of the m. erector spinae. On the other hand, significant group differences in favor of the SOG were observed for psoas major muscle volume (*p* = 0.021, d’: 1.18) and particularly psoas major muscle tissue volume (*p* = 0.001, d’: 1.80).

This finding was unexpected because the m. erector spinae and not the m. psoas major should be predominately addressed by the application of the active spinal orthosis.

Reviewing the available scientific literature, the present finding of missing significant effects at the m. erector spinae was confirmed by some earlier studies. In the Franconian Osteosarcopenia Trial (FROST), an 18-month high intensity resistance exercise (HIT-RT, [14]) program (that included various back-strengthening exercises) had no effect on fatty infiltration of the m. erector spinae [7] in 43 older osteo-sarcopenic men [15]. This finding was not expected because significant effects for LS-BMD [16] and significant effects on muscle fatty infiltration at the mid-femur [17] were observed in this cohort. Berry et al. [18], who applied 10 weeks of back-strengthening exercises using an isokinetic dynamometer, also reported no changes in muscle size or fatty infiltration (T1-weighted images centered on L4) of the lumbar extensors (erector spinae and multifidus muscles) in their cohort of 14 low-back-pain patients. Further, a systematic review (6 studies) [8] on resistance exercise effects on fatty infiltration in paraspinal muscles in people with low back pain (LBP) concluded that “moderate evidence was available that paraspinal fatty infiltration was not reversible in people with LBP”, although one non-controlled exercise trial [19] reported significant favorable changes of m. erector spinae and multifidus percentage fat infiltration after 16 weeks of free-weight-based resistance training. However, results were obtained from MR T2-weighted images that do not provide a quantitative fat infiltration measure. Even more recently, two independent prospective studies showed that fatty infiltration, size, or CT density were not predictive for incident vertebral fractures [20,21]. Unfortunately, none of these studies included the m. psoas major.

This mounting (although still weak) evidence of the insensitivity of the m. erector spinae to physical exercise despite the age-related increase in fat infiltration is difficult to interpret. In general, isometric resistance training has been demonstrated to induce significant maximal force production and hypertrophy, largely regardless of training intensity [22]. Nevertheless, after 16 weeks of twice daily 2–3 h active spinal orthosis application, we observed non-significant effects for maximum isometric back extension strength and m. erector spinae hypertrophy. In contrast and rather unexpectedly, significant effects were noted for maximum trunk flexion strength as well as m. psoas major intra-fascial and muscle tissue volume. Due to different muscle fiber composition, predominately tonic working muscles (e.g., erector spinae and multifidus muscles) revealed a lower potential for exercise-induced hypertrophic effects compared to phasic muscle groups (e.g., m. psoas major) [23]. Nevertheless, we are unable to propose a reasonable explanatory approach for the significant effect of the active spinal orthosis on m. psoas major hypertrophy and trunk flexors strength development. One may speculate that due to pain reduction, participants were more physically active and that this might involve enhanced hip flexion during locomotion. However, only minor increases of physical activities were reported in the SOG, and hence unlikely to result in a roughly 5% increase in psoas major muscle tissue volume. Apart from changes in exercise, we did not observe changes in confounders (e.g., diseases, pharmacologic therapy, injuries) with a potential impact on fatty muscle infiltration in paraspinal muscles. Even if there were specific confounders not addressed by our monitoring, it is difficult to imagine that these variables would influence our measurements of the erector spinae and psoas major muscles differently.

Apart from the low sample size, the present study revealed some other limitations and particularities that should be noted. (1) Due to the large number of missing MRT assessments at follow-up, we refrained from an intention-to-treat analysis (ITT) that is suggested for randomized controlled trials. While most of the participants (50 of 80) were checked by MRI to ensure vertebral fractures or deformities at baseline, only a sample of 21 participants could be acquired for the additional MRI follow-up assessment. Thus, 29 data sets, i.e., about 60% of FU-missing values, must be imputed; correspondingly, imputations and resulting estimates become increasingly speculative. (2) One may argue that significant effects were caused by multiple comparison testing; however, due to the high level of significance for m. psoas major muscle tissue volume (*p* = 0.0007), the results remained statistically significant even after Bonferroni correction, albeit a debatable procedure in this context [12]. (3) We do not define a core outcome for the present assessment since it is still unclear whether IMAT or MT-related parameters, or a combination, may best represent muscle fat infiltration; moreover, this may also depend on the specific muscle. Of note, this aspect may also be seen as an argument in favor of our approach of dispensing with a hypothesis or a sample size analysis for the present research issue. (4) Due to the increased risk of vertebral fracture in this cohort, we opted to apply an isometric assessment of the trunk extensors in an upright position for safety aspects. (5) In this study, we analyzed only six slices per patient instead of the whole volume. An initially intended interpolation between the slices proved to be impractical and manually segmenting additional slices too time-consuming. Nevertheless, the slice-wise approach is frequently used and reported in the literature [19,24,25]. However, as we are dealing with patients with severe curvatures of the spine in some cases, it cannot be ruled out that analyzing the whole volume may lead to smaller errors in the evaluation of the longitudinal results, despite our extensive visual inspection of the segmented regions. (6) In contrast to paraspinal muscles, we have published data on intra- and inter-reader precision for the assessment of muscle and IMAT volume and of fat fraction in the thigh. The % RMSCV values were below 1% [11]. We expect similarly precise values, as segmentation of the paraspinal muscles is easier than that of the thigh. (7) The trial focuses on a low intensity, (very) high volume back-strengthening intervention in women 65 years and older with hyperkyphosis, chronic back pain, and vertebral fractures >3 months. As described, we used the comprehensive study of Hettchen et al. [3] on the effects of the Spinomed active in older kyphotic women with vertebral fracture as a vehicle for the present study. We believe that the present study group is a suitable cohort to address the present research issue. This assessment is based on the reasonable possibility that the paraspinal muscles may have degenerated significantly due to pain and restricted movements in this older female cohort; thus, the clinical relevance of our approach is justified. On the other hand, changes in fatty muscle infiltration cannot necessarily be transferred to younger/more capable and less disabled cohorts without degeneration of the paraspinal musculature. Thus, the generalization of our results to other cohorts (and other interventions) and studies with traditional exercise training is limited.

## 5. Conclusions

Although the present study observed a widely consistent favorable trend with a few significant effects of the active spinal orthosis, one measure of fatty muscle infiltration—evidence for the reversibility of fatty infiltration on the paraspinal muscles—remained vague. Looking at the present findings in the context of the underlying EMSOAT project, we conclude that the pronounced effects on kyphosis and low-back pain cannot be explained by alterations of muscle fatty infiltration of the paraspinal muscles.

## Figures and Tables

**Figure 1 healthcare-13-01262-f001:**
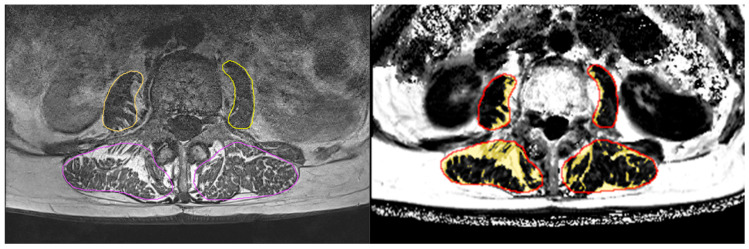
(**Left**) Segmentation of T1w images, with erector muscles in violet and m. psoas major in yellow. (**Right**) IMAT as yellow overlay in Dixon FF image after registration of the segmented muscle regions (red). The results were averaged for the right and left m. psoas major and m. erector spinae.

**Table 1 healthcare-13-01262-t001:** Baseline characteristics of the spine orthosis group (SOG) and the non-treated control group (CG). Mean values (MV) ± standard deviation (SD).

Variable	SOG (*n* = 11) MV ± SD	CG (*n* = 10) MV ± SD	*p*
Age [years]	77.6 ± 5.8	73.3 ± 8.3	0.477
Body height [cm] ^1^	160.4 ± 4.4	162.9 ± 10.4	0.488
Body mass [kg] ^2^	60.1 ± 10.5	59.8 ± 9.7	0.944
Vertebral fractures [number] ^3^	2.45 ± 0.69	2.50 ± 0.85	0.894
Vertebral fracture age [years] ^4^	3.10 ± 1.64	3.56 ± 1.99	0.576
Osteoporosis Medication [*n*] ^3^	8	7	0.890
Kyphosis-angle [°] ^5^	48.6 ± 6.6	46.1 ± 9.0	0.482
Back Pain Intensity [Index] ^6^	2.8 ± 1.2	2.2 ± 1.4	0.305
Back Extensor Strength [kg] ^7^	25.8 ± 7.6	29.6 ± 8.3	0.279

^1^ As assessed by a calibrated stadiometer (Holtain, Crymych Dyfed, Great Britain); ^2^ as assessed by Bio-Impedance Analysis (DSM-BIA, InBody 770, Seoul, South-Korea); ^3^ as determined by medical records or MRT assessment; ^4^ age of the most recent fracture; ^5^ as assessed in straightened upright position using the Debrunner kyphometer (Protek, Bern, Switzerland); ^6^ as assessed by Nominal Rating Score (NRS) from 0 (no pain) to 10 (worst possible pain); ^7^ as determined by a Back-Check^®^ 607, Dr. Wolff, Arnsberg, Germany.

**Table 2 healthcare-13-01262-t002:** Baseline data and changes of parameters related to muscle fatty infiltration in the spine orthosis group (SOG) or the non-treated control group (CG). Mean values (MV) ± standard deviation (SD); 95%-Confidence Interval (95% CI) and significance (*p*) after ANCOVA. Superscript numbers indicate significance (*p*) of changes after 4 months.

Variable	SOG MV ± SD	CG MV ± SD	Difference MV (95% CI)	*p*
**Erector spinae muscle volume [cm^3^]**				
Baseline value	10.4 ± 1.82	10.3 ± 1.43		0.962
Changes after 4 months	0.133 ± 0.282 ^0.153^	0.021 ± 0.295 ^0.829^	0.115 (−0.161 to 0.391)	0.393
**Erector spinae fat fraction [%]**				
Baseline value	34.8 ± 14.1	26.0 ± 6.0		0.084
Changes after 4 months	−0.23 ± 2.22 ^0.734^	0.00 ± 2.32 ^0.999^	0.09 (−2.12 to 2.31)	0.930
**Erector spinae MT volume [cm^3^]**				
Baseline value	6.06 ± 1.89	7.42 ± 1.40		0.086
Changes after 4 months	0.148 ± 0.398 ^0.231^	−0.027 ± 0.417 ^0.835^	0.227 (−0.158 to 0.612)	0.231
**Erector spinae MT fat fraction [%]**				
Baseline value	12.0 ± 3.9	8.8 ± 1.51		0.029
Changes after 4 months	−0.510 ± 1.09 ^0.503^	0.332 ± 1.15 ^0.349^	0.941 (−1.08 to 1.04)	0.096
**Erector spinae IMAT volume [cm^3^]**				
Baseline value	4.31 ± 2.22	2.93 ± 0.091		0.085
Changes after 4 months	−0.064 ± 0.154 ^0.189^	0.040 ± 0.162 ^0.422^	0.084 (−0.068 to 0.237)	0.646
**Psoas major muscle volume [cm^3^]**				
Baseline value	3.89 ± 0.95	3.71 ± 0.55		0.609
Changes after 4 months	0.084 ± 0.177 ^0.257^	−0.156 ± 0.227 **^0.044^**	0.257 (0.004 to 0.470)	**0.021**
**Psoas major fat fraction [%]**				
Baseline value	14.8 ± 3.1	13.2 ± 5.5		0.428
Changes after 4 months	−0.17 ± 1.56 ^0.716^	0.32 ± 1.49 ^0.509^	0.41 (−1.86 to 1.04)	0.561
**Psoas major MT volume [cm^3^]**				
Baseline value	3.30 ± 0.79	3.22 ± 0.62		0.741
Changes after 4 months	0.178 ± 0.171 **^0.003^**	−0.127 ± 0.168 **^0.047^**	0.309 (0.150 to 0.467)	**0.001**
**Psoas major MT fat fraction [%]**				
Baseline value	6.95 ± 0.98	6.25 ± 0.87		0.890
Changes after 4 months	−0.242 ± 1.04 ^0.503^	0.171 ± 1.14 ^0.651^	0.291 (−0.73 to 1.31)	0.428
**Psoas major IMAT volume [cm^3^]**				
Baseline value	0.591 ± 0.251	0.500 ± 0.390		0.293
Changes after 4 months	−0.034 ± 0.094 ^0.253^	−0.044 ± 0.095 ^0.161^	0.019 (−0.066 to 0.104)	0.646

Statistically significant effects are printed in bold.

## Data Availability

The raw data supporting the conclusions of this article will be made available by the authors on request.

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
