# Peer review of "Effects of Active Spinal Orthosis on Fatty Infiltration in Paraspinal Muscles in Kyphotic Women with Osteoporotic Vertebral Fracture—Sub-Analysis of a Randomized Controlled Trial"

_healthcare, 2025, doi:10.3390/healthcare13111262_

Round 1
Reviewer 1 Report
Comments and Suggestions for Authors
Dear Authors. Your effort in conducting the manuscript entitled " Effects of active back orthosis on fatty infiltration in paraspinal muscles in kyphotic women with osteoporotic vertebral fracture – sub-analysis of a randomized controlled trial" is quite obvious. However, the presentation of the study needs some improvements, especially in highlighting the significance of the study. Additionally, some points in the methods section need more clarification and better presentation, such as the outcome measures that should be clearly presented as per the purpose of the study.
I attached an annotated document containing a few comments to be considered.
best wishes

The overall quality of the English is good. Yet, it needs some improvements.
Reviewer 2 Report
Comments and Suggestions for Authors
This sub-analysis investigates the effects of an active back orthosis on fatty infiltration in paraspinal muscles in kyphotic women with osteoporotic vertebral fractures using MRI-based outcomes. The topic is timely and clinically relevant, especially given the aging population and the burden of osteoporosis-related complications. The study is well-designed as an extension of a randomized controlled trial, and the MRI-based assessment adds valuable insight into muscle quality rather than just quantity. However, there are areas needing clarification and improvements, particularly in the introduction, description of the methodology, participant selection, and imaging analysis. Additionally, details on the rationale for conducting this sub-analysis and potential implications could be expanded in the introduction and discussion sections.
Introduction
- Could you elaborate more on the specific pathophysiology of osteoporosis-induced vertebral fractures? How do these fractures typically progress, and what are the key challenges in managing them?
- How does the "Spinomed active" orthosis compare to other types of orthoses or interventions in terms of effectiveness? Are there other studies that provide a comparison of active spinal orthoses versus passive ones, or are you the first to investigate this?
- Why did you choose to focus specifically on fatty infiltration in paraspinal muscles? Are there other muscle parameters (e.g., muscle mass, strength) that might also contribute to the observed benefits of the orthosis? How do these factors interrelate with the clinical outcomes you're studying?
- What specific characteristics of the cohort (e.g., age, severity of osteoporosis, baseline muscle quality) make them ideal candidates for this study? Could the findings be generalized to younger populations or those with less severe conditions?
- Could you provide more detail on how the "Spinomed active" orthosis might influence the muscle parameters you're studying? Does it specifically target muscle recruitment patterns, or does it primarily work by stabilizing the spine and reducing strain?
- MRI is a powerful tool for assessing muscle quality. Could you clarify why MRI was chosen as the method to measure fatty infiltration? Are there any limitations to MRI in this context that you have considered?
- You reference a prior study that assessed the effects of the orthosis on back pain and physical function. How do the findings of this earlier research align with the focus of your current study on muscle infiltration? Do you anticipate similar outcomes in terms of trunk strength and kyphosis?
- What led you to use a subgroup of osteopenic women with vertebral fractures in this study? Could this subgroup limit the generalizability of the results to other populations, such as those with more severe osteoporosis or men?
- In what ways do you envision the findings of this study influencing clinical practice for managing vertebral fractures in older adults? Would you recommend incorporating active spinal orthoses as a standard treatment for these patients?
Materials and methods
Comment:1
The term "semi-blinded" is used to describe the trial. It would help to clarify who exactly was blinded outcome assessors, participants, or data analysts?
Comment:2
The main study included 80 participants, but this sub-analysis includes only 21 who underwent MRI. Please explain the selection criteria for these 21. Were they randomly selected or self-selected? Could this introduce bias?
Comment:3
Also, was any power analysis conducted to determine whether 21 participants are sufficient to detect a meaningful difference in fat infiltration parameters?
Comment:4
The detailed description of the orthosis and compliance monitoring is commendable. However, it might be helpful to clarify how wearing compliance was quantified and whether there was a threshold for “adherence” that participants had to meet.
Comment:5
The use of consistent timing, equipment, and a blinded assessor strengthens internal validity. However, the duration of the assessments (90 minutes) is mentioned in passing — consider elaborating briefly on what this timeframe entailed.
Comment:6
It would be useful to mention whether the assessor performing MRI segmentation was blinded to group allocation.
Comment:7
One concern is the subject-specific thresholding method for muscle fat quantification. Although previously published [10], a brief explanation or a sentence summarizing how this improves accuracy (e.g., versus fixed thresholds) would help the general reader.
Comment:8
Were intra-rater or inter-rater reliability assessments conducted for segmentation?
Comment:9
Ensure consistency in the terms used for paraspinal muscle groups (e.g., always refer to "psoas major" and "erector spinae" using full terms initially, then use abbreviations if needed).
Comment:10
In the imaging section, clarify whether FF refers to the fat fraction of the total muscle volume or a specific sub-region (e.g., within MT only).
Comment:11
Typo in line: "Invention The intervention utilized…" should be corrected.
Consider standardizing unit presentations (e.g., always use "mm³" and "cm" consistently).
Comment:12
Please specify which statistical software and version (e.g., SPSS v26, R v4.3.0, or others) was used to perform the analysis.
Reviewer 3 Report
Comments and Suggestions for Authors
Dear authors,
We have received your manuscript entitled:
“Effects of Active Back Orthosis on Fatty Infiltration in Paraspinal Muscles in Kyphotic Women with Osteoporotic Vertebral Fractures – Sub-analysis of a Randomized Controlled Trial.”
Your study explores whether wearing an active spinal orthosis can improve paraspinal muscle quality in older women with osteoporotic vertebral fractures. This is an important topic, and your use of MRI to measure fatty infiltration and muscle volume is a strength. The randomized controlled trial design also supports the reliability of your results.
Your manuscript is of interest, but a revision is required before it can be considered for publication. Please address the following points:
Clarify the MRI analysis methods: Please state whether MRI segmentation was performed by a blinded assessor. Also explain why only six axial slices were analyzed instead of the full muscle volume, and discuss how this may have affected the results.
Improve the results section: Be careful not to overstate the findings. Clearly explain that only psoas muscle volume and tissue volume showed significant improvement, while no changes were found in fatty infiltration or in the erector spinae.
Strengthen the discussion of unexpected findings: The fact that psoas muscles improved while erector spinae did not is unexpected. Please discuss possible reasons for this result more thoroughly, and clearly state that this was not anticipated.
Revise the abstract and conclusion: These sections should match the results and emphasize that significant changes were only seen in the psoas, with no effect on fatty infiltration or on the expected target muscle (erector spinae). Avoid general statements like “positive effects on fatty infiltration.”
Correct formatting and language issues: The manuscript contains several typographical and formatting errors (e.g. “angel” instead of “angle”). Please revise the text for clarity, spelling, and consistency.
Once these revisions have been completed, we would be glad to reconsider your submission.
Best regards
Round 2
Reviewer 2 Report
Comments and Suggestions for Authors
Dear Authors, Thank you for your revised submission of the manuscript titled "Effects of active spinal orthosis on fatty infiltration in paraspinal muscles in kyphotic women with osteoporotic vertebral fracture – sub-analysis of a randomized controlled trial". I have carefully reviewed the revised version and I am pleased to inform you that you have satisfactorily addressed all my previous comments and suggestions. Thank You